# Identification and Characterization of Circular RNAs Involved in the Flower Development and Senescence of *Rhododendron delavayi* *Franch*

**DOI:** 10.3390/ijms231911214

**Published:** 2022-09-23

**Authors:** Xiaorong Xu, Yufeng Xiao, Ximin Zhang, Ming Tang, Jing Tang

**Affiliations:** 1Key Laboratory of State Forestry Administration on Biodiversity Conservation in Karst Mountainous Areas of Southwestern China, School of Life Sciences, Guizhou Normal University, Guiyang 550001, China; 2State Key Laboratory of Plant Physiology and Development in Guizhou Province, Guizhou Normal University, Guiyang 550001, China; 3Key Laboratory of Plant Physiology and Development Regulation, Guizhou Normal University, Guiyang 550001, China

**Keywords:** circRNA, flower senescence, *Rhododendron delavayi Franch*

## Abstract

Floral development and senescence are a crucial determinant for economic and ornamental value. CircRNAs play an essential role in regulating plant growth and development; however, there is no systematic identification of circRNAs during the lifespan of flowers. This study aims to explore the expression profile and functional role of circRNAs in the full flowering stages of *Rhododendron delavayi Franch*. We carried out transcriptome sequencing of the six stages of *Rhododendron delavayi Franch* flowers to identify the circular RNA expression profile. In addition, using bioinformatics methods, we explored the functions of circRNAs, including analysis of the circRNA-miRNA-mRNA network, short time-series expression miner (STEM), and so on. We identified 146 circRNAs, of which 79 were differentially expressed from the budding to fading stages. Furthermore, using STEM analysis, one of the 42 circRNA expression model profiles was significantly upregulated during the senescence stage, including 16 circRNAs. Additionally, 7 circRNA-miRNA-mRNA networks were constructed with 10 differentially expressed circRNAs, in which some target mRNA may regulate the development and senescence of the Rhododendron flowers. Finally, by analyzing the correlation between circRNAs and mRNA, combined with existing reports, we proposed that circRNAs play a regulatory role during flower development and senescence by mediating the jasmonate signaling pathway. Overall, these results provide new clues to the potential mechanism of circRNAs acting as novel post-transcriptional regulators in the development and senescence process of flowers.

## 1. Introduction

Flowers are important reproductive organs for plants and are also of important ornamental values. *Rhododendron delavayi Franch* belongs to the genus Rhododendron and is a highly attractive perennial ornamental tree [1] with beautiful and brightly colored flowers. Flowers undergo a series of developmental, physiological, and metabolic changes in an orderly manner. Senescence is a fundamental developmental process involving complex regulation at multiple levels [2]. Moreover, development and senescence have a significant influence on the length of flower lifespans of plants, which is of great significance to the ornamental value of flowers.

CircRNAs are a type of non-coding RNAs with a covalently closed loop structure. They do not possess 3′ polyadenylic acid tails (polyA) and 5′ caps, and have by-products ubiquitously produced in the reverse splicing process in eukaryotes [3,4]. CircRNAs were discovered between the 1980s and 1990s [5,6,7]. At that time, only a small number of circRNAs were found. Thus, circRNAs were considered as by-products of the splicing process and were thought to have no definitive functions [8]. Although how circRNAs are generated is still unclear, circRNAs can be divided into three categories, according to their production source: exonic circRNAs; intronic circRNAs; and intergenic circRNAs [1]. With the rapid development of high-throughput sequencing technology, circRNAs in many different cell types or tissues, and of many species, have been sequenced in a whole-genome manner; thus, more and more circRNAs have been discovered and identified [9,10]. Previous studies have shown that circRNAs have important post-transcriptional regulatory factors [11]. For example, in humans, zebrafish, mice, and in many other model animals, some circRNAs were found to be highly expressed at specific developmental stages [11,12,13]. In addition, previous studies have shown that some circRNAs contain miRNA response elements (MREs), which can bind to miRNAs, thereby isolating and preventing the miRNAs from binding to the corresponding target genes. In this process, these circRNAs act as miRNA sponges and prevent miRNAs from inhibiting the target genes, thus regulating the gene expression [14]. Since circRNAs have been found to be involved in the growth and developmental regulation in multiple animals, it is expected that circRNAs may also exhibit different functions and expression patterns during flower development and senescence.

Although it has been confirmed that circRNAs play an indispensable role in regulating animal growth and development, research on the characterization and function of plant circRNAs is still in its preliminary stage. In 2014, plant circRNAs were first discovered in *Arabidopsis thaliana* [15]. Later, circRNAs were identified in more and more plants and were found to play critical roles in many biological processes. A total of 6612 circRNAs in rice were identified, and it was found that circRNAs regulate many biological processes in plants, such as hormone signaling, proteolysis, DNA damage repair, leaf senescence, and through regulation of gene expression [8]. For instance, overexpression of a rice circRNA, CircR5g05160, can reduce the expression level of the LOC_Os05g05160 gene [16]. Tong et al. found that circRNAs are involved in the regulation of photosynthesis and metabolite biosynthesis during the development of *Camellia sinensis* leaves [17]. Liu et al. constructed a circRNA-miRNA-mRNA network to study the function of circRNAs during leaf development and senescence of *Arabidopsis thaliana*. Their analysis showed that circRNAs might be involved in phytohormone signal transduction and porphyrin/chlorophyll metabolism during leaf senescence [18]. Zuo et al. constructed a lncRNA-circRNA-mRNA network studying cold stress in sweet peppers and confirmed the circRNAs function in cold injury response [19]. Recent studies have shown that circRNAs play an essential role in the leaf senescence process of rice and *Arabidopsis*. Comparative analysis of high-throughput sequencing data of circRNAs in the soybean cytoplasmic male sterile line, and its maintainer line, showed that circRNAs might be involved in the development regulation of flowers and pollen [20]. However, there is no report on the role of circRNAs in the flower development and senescence of *Rhododendron delavayi Franch*.

To identify circRNAs involved in flower development and senescence of *Rhododendron delavayi Franch* and to explore their potential regulatory functions, we carried out high-throughput transcriptome sequencing of flowers at different stages. With the sequencing data, we identified a comprehensive list of circRNAs related to flower development. Some of these circRNAs are related to the flower senescence process of *Rhododendron delavayi Franch*. We also inferred the potential role of these circRNAs using multiple methods, including time-series profile analysis, circRNA-miRNA-mRNA network construction, and gene enrichment analysis.

## 2. Results

### 2.1. Identification of circRNAs in the Rhododendron delavayi Franch Flowers

To systematically identify the circRNAs related to the senescence process of *Rhododendron delavayi Franch* flower, the flowers at the bud stage (TB), pollen dehiscence stage (TPD), pollination stage (TP), after pollination stage (TAP), senescence stage (TS), and the fading stage (TF), were sampled with 3 biological repeats in each stage (Figure 1). In total, 18 RNA libraries were constructed. The Illumina HiSeq 3000 platform was used for high-throughput transcriptome sequencing, yielding a total of nearly 120 G raw sequencing data with Q20 greater than 95%, Q30 greater than 90%, and GC content at 45.5–47.5% (Appendix A). After data cleaning, the Trinity software was used for transcriptome assembly.

The Cirit package was used for circRNA identification based on the assembled transcriptome. In total, 146 circRNAs were identified in the *Rhododendron delavayi Franch* flower samples. Among these, 98, 94, 108, 107, 123, and 114 circRNAs were identified in the TB, TPD, TP, TAP, TS, and TF stages, respectively. The TS stage possessed the highest number (123, 84.2%) of circRNAs, followed by the TF stage (114, 78.1%) (Figure 2a). The result suggested that a large number of circRNAs may be involved in the regulation of Rhododendron flower senescence. Sixty-one circRNAs (41.8%) were found in all six flowering stages (Figure 2b), suggesting that more circRNAs are uniquely expressed during the developmental and senescence stages of flower. Then, using the BLAST circRNA annotation function of the PlantCircNet database, we found only 4 circRNAs in the flowers of *Rhododendron delavayi Franch* that were of high similarity to the 13 circRNAs in the *Arabidopsis*, barley, and soybeans (the average similarity ≥ 83.5%). These results are consistent with previous reports indicating that circRNAs were specifically expressed in different plant developmental stages [21]. Moreover, the functions of many circRNAs are unclear in the *Rhododendron delavayi Franch* flower.

### 2.2. Analysis of Differentially Expressed circRNAs

To investigate the expression pattern of circRNAs from budding to fading in the flower development of *Rhododendron delavayi Franch*, we analyzed the expression quantity and identified many differentially expressed circRNAs (DEcircRNAs) between all pairs of stages based on the chronological order (i.e., TB vs. TPD; TPD vs. TP; TP vs. TAP; TAP vs. TS; TS vs. TF).

Considering only circRNAs expressed in at least two biological replicates, we performed differential analysis of 146 circRNAs identified in the six developmental stages. A comparative analysis of all pairs of stages identified 28 (TB vs. TPD), 30 (TPD vs. TP), 30 (TP vs. TAP), 45 (TAP vs. TS), and 40 (TS vs. TF) DEcircRNAs, respectively, (*p* ≤ 0.05). In total, 79 DEcircRNAs at different developmental stages (Appendix A) were obtained. Compared with the earlier stages, the most DEcircRNAs were in the senescence stage (79.7%). Since these DEcircRNAs are derived from protein-coding genes, we further analyzed whether there was a correlation between DEcircRNAs and the transcript of the host gene. We found that DEcircRNAs during the developmental and senescence stages of flower are not the results of the change transcript of the corresponding host genes. These results further support that circRNAs may play an important roles in the development and senescence of Rhododendron flowers. Then, the expression of DEcircRNAs during the process of Rhododendron flower developmental to senescence was analyzed (Figure 3). A large portion of DEcircRNA only upregulated at a specific time point, especially in the senescence and fading stages (33, 41.8%). These results suggested that circRNAs may play important roles during the developmental and senescence stages of *Rhododendron*
*delavayi Franch* flower, especially in the senescence stages.

### 2.3. Time Series Profile Analysis of Transcript Expression

To further analyze the expression pattern of circRNAs, we used the Short Time-series Expression Miner (STEM) method. Using STEM analysis, 42 model profiles were established to illustrate the expression patterns of circRNAs. Only one of these 42 model profiles was identified as significantly enriched in the senescence stage (*p* ≤ 0.05) (Figure 4a), including 16 circRNAs (Figure 4b). The expression of these 16 circRNAs was significantly upregulated in the senescence stage, compared with other stages. The result indicated these circRNAs may play an essential role in the senescence process of *Rhododendron delavayi Franch*.

### 2.4. Construction of circRNA-miRNA-mRNA Networks

Some results indicated that circRNAs competitively bind to miRNAs, and subsequently regulate their target genes by acting as miRNA sponges [22,23,24,25,26]. To explore the sponge function of circRNAs to miRNAs, we used the RegRNA2.0 to analyze the potential role of the 79 DEcircRNAs. A total of 10 DEcircRNAs were found that could potentially bind to 7 miRNAs, namely, miR828, miR395, miR833-5p, miR829.1, miR845a, miR3440b-3p, and miR159c (Appendix A). To construct the circRNA-miRNA-mRNA networks related to the development and senescence of *Rhododendron*
*delavayi Franch* flowers, the targets of the 7 miRNAs and 10 DEcircRNAs were predicted using three different tools (psRNATarget, TAPIR, PlantCircNet). Based on that prediction, there were 59 target mRNAs for miR828, 75 target mRNAs for miR395, 63 target mRNAs for miR833-5p, 228 target mRNAs for miR829.1, 289 target mRNAs for miR845a, 95 target mRNAs for miR3440b-3p, and 82 target mRNAs for miR159c. In the transcriptome data of Rhododendron flower, 9 target mRNAs for miR828, 18 target mRNAs for miR395, 12 target mRNAs for miR833-5p, 45 target mRNAs for miR829.1, 57 target mRNAs for miR845a, 21 target mRNAs for miR3440b-3p, and 15 target mRNAs for miR159c, were found. Then, the Cytoscape software was used for the construction of 7 circRNA-miRNA-mRNA networks (Figure 5). These networks showed the connections between DEcircRNAs, miRNAs, and mRNAs, during the process of development to senescence in Rhododendron flowers. Among the obtained mRNA, some of them are related to flower development and senescence in other plants. Some studies found that the mRNA of Cullin 1 (*CUL1*), anaphase promoting complex 10 (*APC10*), cytochrome P450, family 704, subfamily B, polypeptide 1 (*CYP704B1*), and CONSTANS (*CO*) are related to flower development and senescence [27,28,29,30]. The results suggested that circRNAs may be involved in mRNA expression regulation during the development and senescence of Rhododendron flowers.

### 2.5. Functional Annotations of the mRNAs

To explore the function of predicted mRNAs of the DEcircRNAs, the KEGG pathway analysis was performed by Kobas3.0. The result showed that a total of 17 significantly enriched pathways were identified (Figure 6). The pathway includes a variety of metabolic processes, such as beta-Alanine metabolism, Alanine, aspartate and glutamate metabolism, phosphonate and phosphinate metabolism, and so on. In addition, a variety of biosynthesis processes are also involved, such as steroid biosynthesis, phenylalanine, tyrosine and tryptophan biosynthesis; brassinosteroid biosynthesis were also enriched. These results showed that the target mRNA of DEcircRNAs during flower development and senescence were related to various biological processes. It is reported that some of these enriched KEGG pathways are involved in plant development and senescence. For example, the ubiquitin mediated proteolysis pathway related genes included cullin 1 (*CUL1*) and anaphase promoting complex 10 (*APC10*).

## 3. Discussion

Many circRNAs were discovered and identified in human [11], rat [13], and zebrafish [12], and were shown to have important roles in various biological and developmental processes. In contrast to the comprehensive study of circRNAs in animals, circRNAs research in plants is still in its infancy [31]. It was not until the first discovery of circRNAs in *Arabidopsis thaliana* in 2014 [15] that research on circRNA in plants began. Since then, a large number of circRNAs have been identified in various plants, such as cucumber [32], corn [33], wheat [34], rice [2], and sweet pepper [19]. These studies revealed the essential roles of circRNAs in various biological processes in plants. In addition, previous studies have shown that circRNAs can be specifically expressed at different developmental stages [18]. For example, 252 circRNAs that are differentially expressed during fruit development were identified in sea buckthorn [35], and 113 circRNAs were identified in the flag leaf development of senescence processes in rice [18]. In our research, 146 circRNAs were identified, of which 79 circRNAs were differentially expressed in the six flower developmental stages of *Rhododendron delavayi Franch*. Furthermore, a large portion of differentially expressed circRNA only upregulated at a specific time point, especially in the senescence and fading stages. In addition, the upregulated DEcircRNAs are not the results of the change transcript of corresponding host genes.

Meanwhile, we found that the circRNAs identified from Rhododendron flowers also exhibited a development-specific expression pattern. Some circRNAs were only expressed in specific developmental stages. For example, Rd-circR132 was only expressed in the pollination stage of flower development, Rd-circR2, Rd-circR4, Rd-circR5, Rd-circR6, Rd-circR7, Rd-circR18, Rd-circR144, and Rd-circR145, were only expressed in the senescence stage of flower development, and Rd-circR3, Rd-circR8, and Rd-circR114, were only expressed in the late stage of flower development. CircRNAs specifically expressed in a developmental stage may play a role in that specific stage. In addition, with STEM analysis to explore the expression patterns of circRNAs in the six flower development stages of *Rhododendron delavayi Franch*, one model profile was significantly upregulated in the senescence stage. Therefore, these results indicated that circRNAs might act as important functional regulators in the flower development and senescence process of *Rhododendron delavayi Franch*.

More and more studies have shown that circRNAs act as sponges for miRNAs and play a regulatory role through the circR-NA-miRNA-mRNA network. For example, it has been reported that 53 differentially expressed circRNAs in the circRNA-miRNA-mRNA network of sea-buckthorn fruits might function as the miRNA sponge in fruits development [35]. In another study of circRNAs in rice flag leaves, three networks were constructed and several key circRNAs were found to act as the miRNA sponge, which is involved in large-scale gene regulation related to leaf senescence [2]. In this study, we identified 10 differentially expressed circRNAs that can potentially bind to miRNAs, and constructed 7 circRNA-miRNA-mRNA networks. Firstly, we found that some predicted target mRNA in the network may be involved in the regulation of the development and senescence in Rhododendron flowers. For example, *CYP704B1* is a long-chain fatty acid hydroxylase, which is essential for sporopollenin synthesis and pollen wall formation [29]. The low level of *CYP704B1* expression may lead to insufficient sporopollenin synthesis and pollen wall defects [36]. Our results showed that the expression of Rd-circR83 and Rd-circR103, which regulate *CYP704B1*, was significantly upregulated during the pollen stage. Consistently, *CYP704B1* was also significantly upregulated during the pollen period. Additionally, we found that some of the biological processes enriched by predicted mRNA were associated with development and senescence process to a certain extent, further suggesting that circRNAs may play a potential role in the development and senescence stage. For example, the pentose phosphate pathway and the fructose and mannose metabolic pathway were closely related to control flowering [37,38]. Finally, by analyzing the correlation between circRNAs and mRNA, combined with existing reports, we proposed that circRNAs play a regulatory role during flower development and senescence by mediating the jasmonate signaling pathway (JA signaling pathway) (Figure 7). In the circRNA-miRNA-mRNA pairs, miRNAs showed a negative correlation with corresponding targets circRNAs and mRNAs, while circRNAs showed a positive correlation with mRNAs. The Rd-circR67, which was predicted to regulate *CUL1*, had a significant positive correlation with the transcript level of *CUL1* (annotated as the Cluster-40952.16564 in the transcriptome data) (*p* ≤ 0.05) (Figure 8). *COI1* (CORONATINE INSENSITIVE 1) associates with *CUL1* and ASK (Skp1-like proteins) to assemble ubiquitin-ligase complexes, which we have designated S-phase kinase-associated protein1/Cullin1/F-box protein *COI1* (SCF^COI1^) complexe, which plays a key role in flower development, especially in flowering time and pollen fertility [39,40,41]. We found that *CUL1* had a significant positive correlation with the *COI1* in the transcript data (annotated as the Cluster-40952.41096, Cluster-40952.28573, Cluster-40952.39215 and Cluster-40952.47013) (*p* ≤ 0.05) (Figure 8). During the process of development to senescence in *Rhododendron delavayi Franch* flower, *CUL1* and *COI1* showed significantly higher expression at the stage of pollination and fading. JA receptor *COI1* is involved in regulation of floral transition. Some evidence indicated that the mutation and reduction of *COI1* lead to male sterility and delayed filament elongation, and anther dehiscence. In addition, the mutation and reduction of *COI1* also delayed the petaline abscission [41,42]. Therefore, we speculated that Rd-circR67 participated in the regulation of *CUL1* expression through the adsorption of miR159c, and guaranteed the formation of SCF^COI1^ in the stage of pollination and fading. This mechanism regulated the development and decline of Rhododendron flowers through the JA pathway. These analyses bring new targets for the molecular mechanism of circRNAs involved in the regulation of plant growth and development.

In addition, previous studies have shown that circRNAs have highly conserved sequences in humans and animals [35]. Conservative sequence analysis of soybean circRNAs showed that 551 gene pairs among *Glycine max*, *Oryza sativa,* and *Arabidopsis thaliana* were orthologs, suggesting that circRNAs sequences are also conserved in plants [43]. We further identified four circRNAs in *Rhododendron delavayi Franch* flowers that showed sequence similarity to 13 known plant circRNAs in the PlantCircNet database. These observations indicated that there might be abundant circRNAs with unknown functions in *Rhododendron delavayi Franch,* as well as in other plants. The results maybe present a new challenge for circRNAs in Rhododendrons.

Overall, these results indicate that circRNAs are involved in the regulation of flower development and senescence. Our findings will provide new clues to further explore the molecular regulatory mechanisms of circrnas in flower development and senescence.

## 4. Materials and Methods

### 4.1. Plant Sample Collection

The test materials were *Rhododendron delavayi Franch* flowers at different developmental stages, originating from the Baili Rhododendron Scenic Area, Guizhou Province, China. For *Rhododendron delavayi Franch* of similar growth condition, flowers at the time of bud stage (TB), pollen dehiscence stage (TPD), pollination stage (TP), after pollination stage (TAP), senescence stage (TS), and fading stage (TF), were sampled. For each developmental stage, 3 biological replicate samples were collected. The samples were quickly frozen in liquid nitrogen and then stored in a −80 °C refrigerator for later use.

### 4.2. High Throughput Sequencing and Identification of circRNAs

Total RNAs were extracted and 1.5 μg RNA of each sample was used for library construction. The sequencing libraries were constructed using the NEBNext UltraTM RNA Library Prep Kit and sequenced on the Illumina HiSeq 3000 platform. Clean reads were obtained from raw reads by removing the reads containing adapter sequence and ploy-N, or low-quality reads. The Trinity software [44] was used for the transcriptome assembly, followed by circRNA identification using the Cirit package [45].

### 4.3. Identification of Differentially Expressed circRNAs (DEcircRNAs)

The expression levels of circRNAs were determined by the number of normalized reads. The DEcircRNAs of two flower developmental stages were analyzed using the DESeq2 [46] R package 4.0.4 (R Foundation, Beijing, China) (*p* ≤ 0.05). All detected DEcircRNAs were aligned against the PlantCircNet (http://bis.zju.edu.cn/plantcircnet/index.php, (accessed on 5 January 2021)) database [47] using the BLASTN tool. The BLAST method was used to identify conserved circRNAs using our identified DEcircRNAs as the input.

### 4.4. Time Series Profile Analysis of the Transcriptome

The read counts of the circRNAs were converted into FPKM (expected number of Fragments Per Kilobase of transcript sequence per Millions of base pairs sequenced). Next, the STEM clustering method was utilized to analyze the expression trend of circRNAs from the budding to the fading of *Rhododendron delavayi Franch* [48,49]. For the STEM analysis, default parameters were used. Each circRNA was assigned to a model profile to which its time series was most closely related, based on the correlation coefficient. The significance levels, at which the number of circRNAs assigned to a model profile compared to the expected number of circRNAs assigned, were calculated for each model profile. Profiles with *p* ≤ 0.05 were considered as significant temporal expression profiles [50].

### 4.5. Construction of circRNA-miRNA-mRNA Networks

To predict whether any circRNAs act as miRNA sponges and bind to miRNAs, the RegRNA 2.0 (http://regrna2.mbc.nctu.edu.tw/, (accessed on 23 November 2020)) [51] website was used. To identify potential miRNA targets, the following 3 online target gene prediction tools were used: (1) psRNATarget (https://www.zhaolab.org/psRNATarget/, (accessed on 1 June 2021)) [52], the default parameters of the Schema v2 (2017 release) scoring model were used, a score ≤5 indicates a potential target gene; (2) TAPIR (http://bioinformatics.psb.ugent.be/webtools/tapir. (accessed on 3 June 2021)) [53], default parameters were used, a score ≤4 and a free energy ratio ≤0.7 indicate a potential target gene; and (3) PlantCircNet (http://bis.zju.edu.cn/plantcircnet/index.php, ( accessed on 13 May 2021)) [46], the basic search function with default parameters was used for target gene prediction. The Cytoscape 3.8.2 (Free Software Foundation, Boston, MA, USA) [54] tool was used to draw a network diagram of potential relationships between circRNAs, miRNAs, and mRNAs.

### 4.6. Functional Annotations

The Kobas 3.0 (http://kobas.cbi.pku.edu.cn/, (accessed on 26 May 2022)) website was used to perform the KEGG pathway enrichment analysis on the predicted target mRNAs. A pathway with *p* ≤ 0.05 was considered to be significantly enriched.

### 4.7. Correlation Analysis

The IBM SPSS Statistics 20 (IBM Corp., Armonk, NY, USA) software was used to analyze the Pearson correlation coefficient between the circRNAs and predicted target mRNAs.

## 5. Conclusions

In summary, we identified 146 circRNAs in the six developmental stages of *Rhododendron delavayi Franch* flowers. An analysis of the expression patterns of circRNAs and the circRNA-miRNA-mRNA networks showed that circRNAs participate in the flower development of *Rhododendron delavayi Franch* via the carbohydrate metabolic pathway, photosynthetic carbon fixation pathway, and ubiquitin-mediated proteolytic pathway. CircRNAs maybe play a regulatory role in flower development and senescence by positively regulating the CUL1 gene and the mediating jasmonate signaling pathway. Our research revealed that circRNAs might function as novel post-transcriptional regulators in the flower development of *Rhododendron delavayi Franch*. In addition, these results lay the foundation for investigating the response of circRNAs to floral development and senescence.

## Figures and Tables

**Figure 1 ijms-23-11214-f001:**
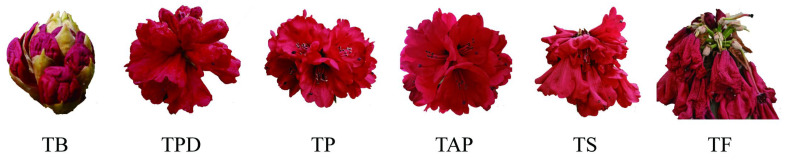
The phenotype of samples in each period. TB: the time of bud stage; TPD: pollen dehiscence stage; TP: pollination stage; TAP: after pollination stage; TS: senescence stage; and TF: fading stage.

**Figure 2 ijms-23-11214-f002:**
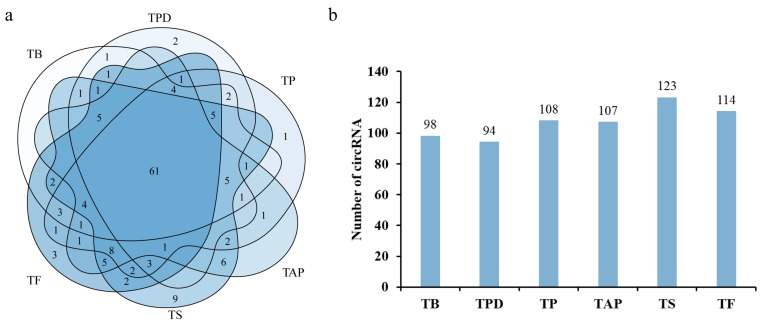
The distribution of circRNAs in different stages: (**a**) a Venn diagram showing the number and distribution of detected circRNAs in six stages; (**b**) histogram shows the number of circRNAs detected in each stage. TB: the time of bud stage; TPD: pollen dehiscence stage; TP: pollination stage; TAP: after pollina-tion stage; TS: senescence stage; and TF: fading stage.

**Figure 3 ijms-23-11214-f003:**
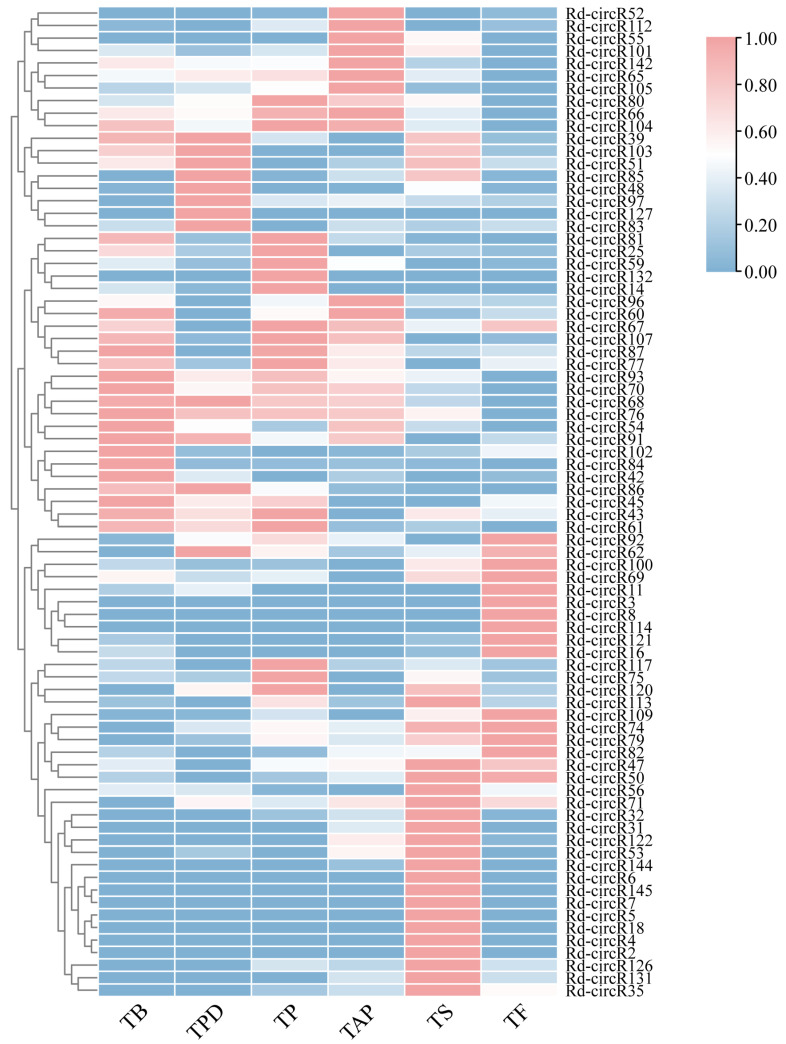
Heatmap showing the expression patterns of 79 DE-circRNAs during the lifespan of flowers. TB: the time of bud stage; TPD: pollen dehiscence stage; TP: pollination stage; TAP: after pollina-tion stage; TS: senescence stage; and TF: fading stage.

**Figure 4 ijms-23-11214-f004:**
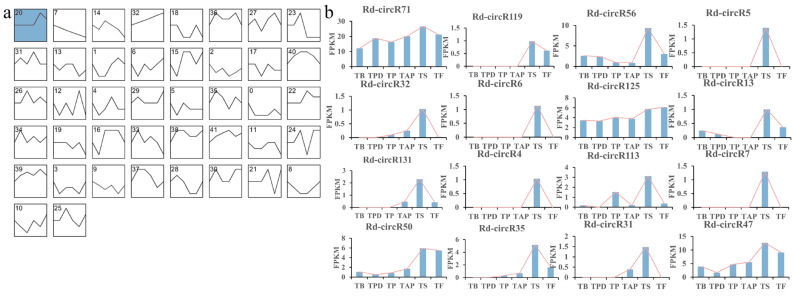
Patterns of circRNA expression in six stages inferred by STEM analysis: (**a**) 42 candidate profiles were obtained via STEM analysis. Only one colored profile is significant profile (*p* ≤ 0.05); (**b**) 16 circRNAs were clustered into this significant expression profile. TB: the time of bud stage; TPD: pollen dehiscence stage; TP: pollination stage; TAP: after pollina-tion stage; TS: senescence stage; and TF: fading stage.

**Figure 5 ijms-23-11214-f005:**
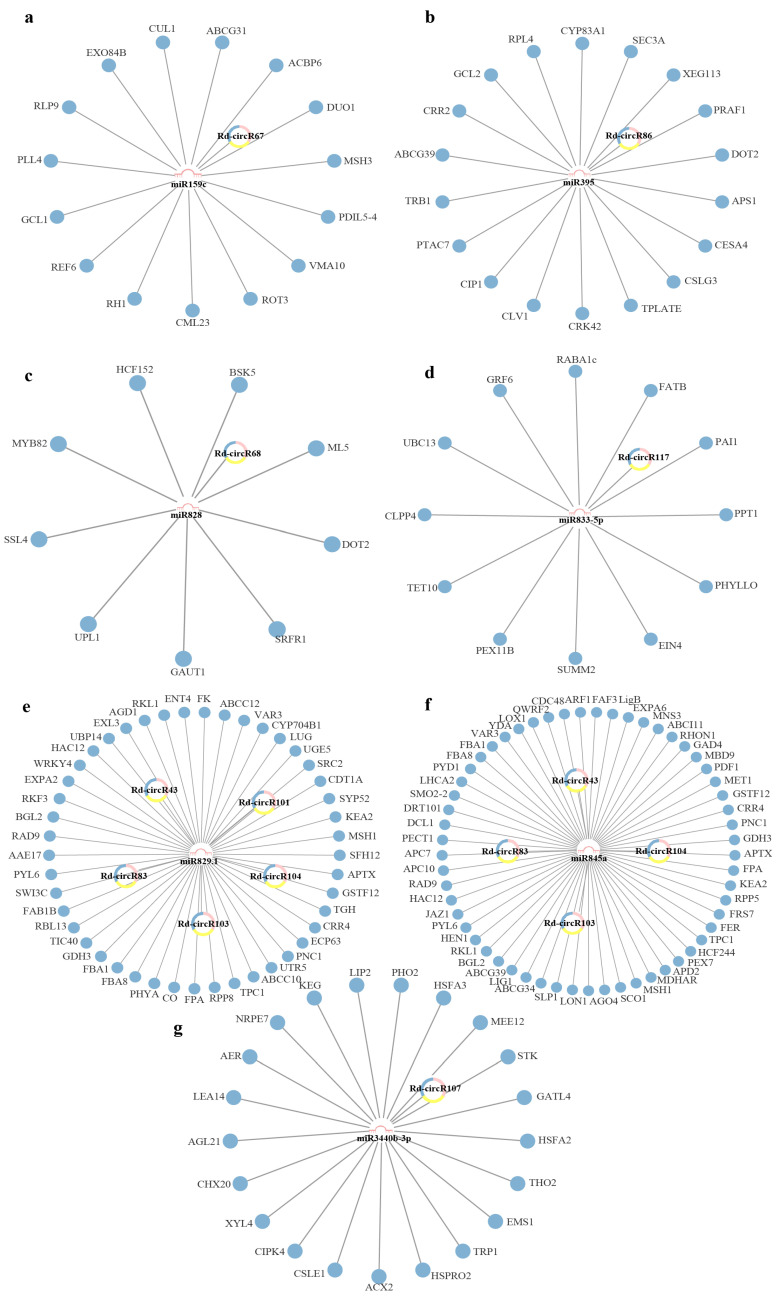
The prediction of circRNA-associated interaction networks during the lifespan of flowers: (**a**–**g**) the circRNA-miRNA- mRNA networks of all the identified circRNAs.

**Figure 6 ijms-23-11214-f006:**
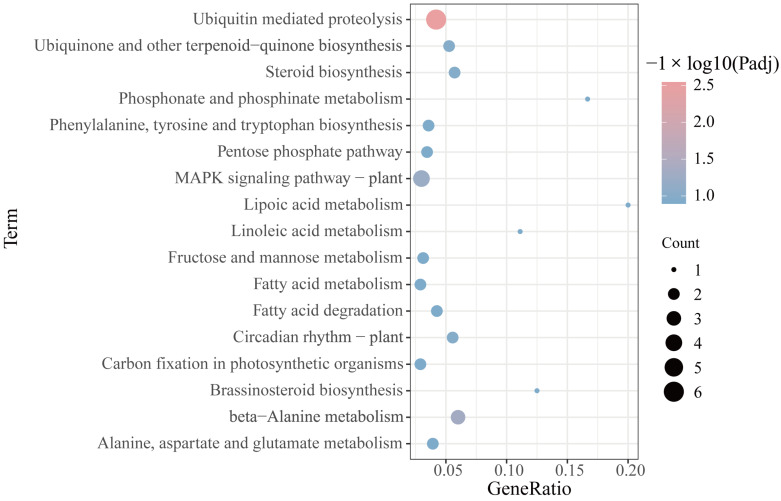
KEGG pathway analysis of mRNA. Only the terms with *p*-value ≤ 0.05 are shown.

**Figure 7 ijms-23-11214-f007:**
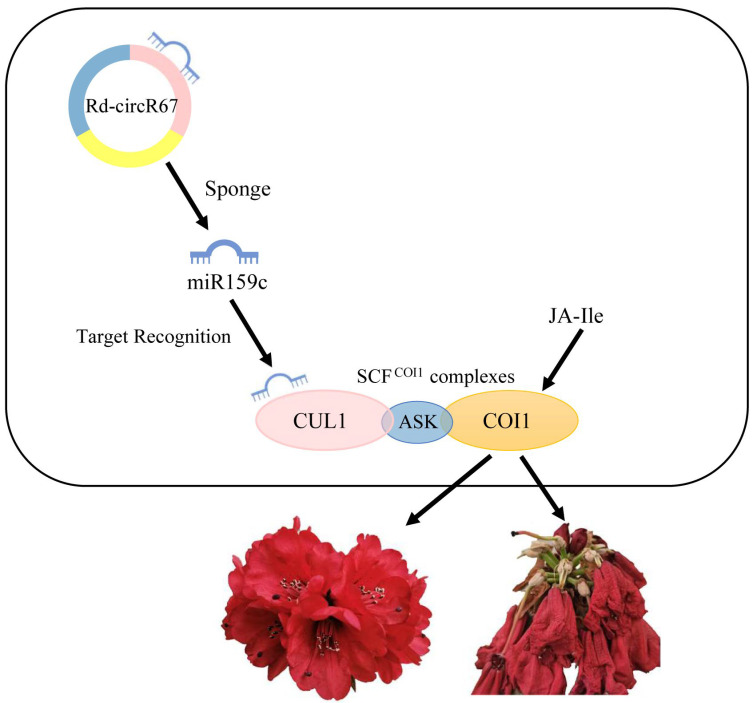
A proposed model for the regulatory network of cicRNA during flower development of Rhododendron floral development and senescence.

**Figure 8 ijms-23-11214-f008:**
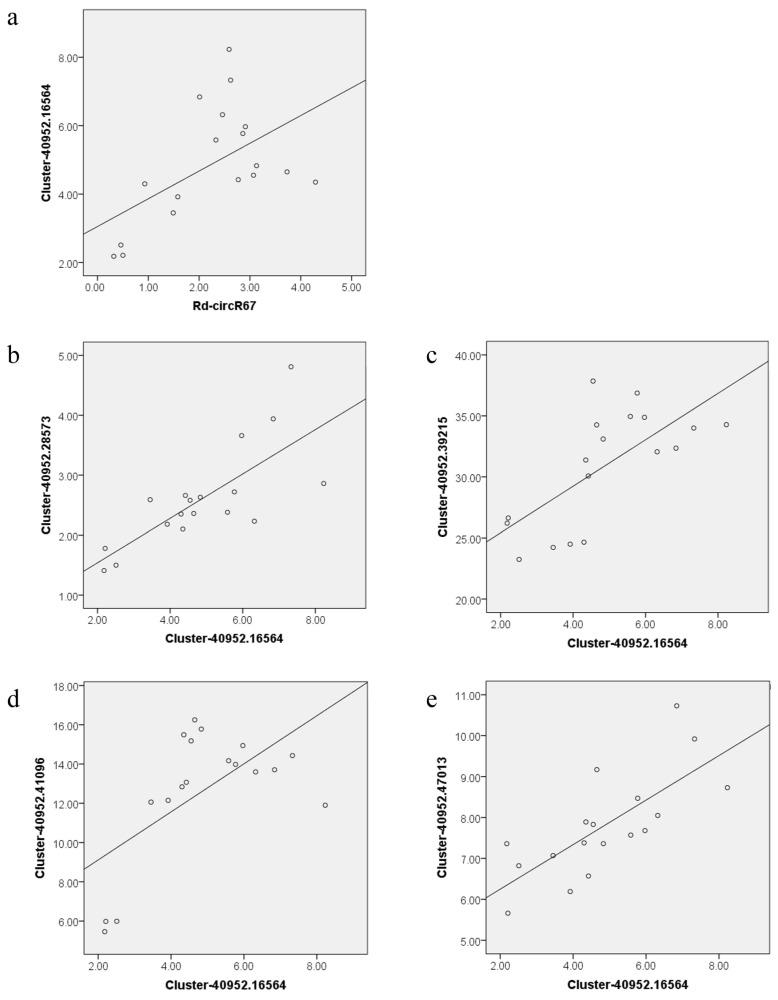
The Pearson correlation coefficient between the circRNAs and predicted target mRNAs: (**a**) the Pearson correlation coefficient between Rd-circR67 and *CUL1*; and (**b**–**e**) the Pearson correlation coefficient between *CUL1* and *COI1*.

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
