# Peer review of "Identification and Characterization of Circular RNAs Involved in the Flower Development and Senescence of Rhododendron delavayi Franch"

_ijms, 2022, doi:10.3390/ijms231911214_

Round 1
Reviewer 1 Report
Major points:
1. There are no phenotype pictures to show Rhododendron flowers at the bud stage (TB), pollen dehiscence stage (TPD), pollination stage (TP), after pollination stage (TAP), senescence stage (TS) and fading stage (TF). I think this is very important for people to know what exactly the flowers are like during these stages.
2. There are no any qPCR data to verify the sequencing results. I think this is also very important because usually there are so many false positive results in sequencing data.
3. In their model, the authors conclude that Rd-circR67/miR159c/CUL1 network regulate the development and decline of Rhododendron flower through the JA pathway. But they do not have any phenotype data. I do not trust that this pathway indeed can regulate the development and decline of Rhododendron flower. Unless they can show that by knowing down or overexpressing of these genes, especially Rd-circR67, indeed have some flower development and senescence phenotype.
Minor points:
1. For Figure 1a, it looks like particularly ugly. The authors should make a newer and clearer figure.
2. For Figure 3b, it looks like invisibility. The authors should also make a newer and clearer figure.
3. From line 320-338, The authors should use some figures of analysis data to show the results, rather than only described in words.
Author Response
Dear reviewer:
Thank you for your comments on our manuscript entitled “Identification and Characterization of Circular RNAs Involved in the Flower Development and Senescence of Rhododendron delavayi Franch” (ijms-1893749). According to your suggestions, we have made amendments in the revised manuscript carefully. Here, we respond to your comments point-by-point. Please see the attachment.
Best regards,
Jing Tang, PhD

Reviewer 2 Report
The manuscript Identification and Characterization of Circular RNAs Involved in the Flower Development and Senescence of Rhododendron delavayi Franch could be accepted.
The abstract is too long and needs to focus on the main finding.
The last paragraph needs to justify the aims of the manuscript.
Add conclusion part.
Author Response

(The authors gave the same response as above.)

Reviewer 3 Report
The authors conducted a study on the identification and characterization of circular RNAs involved in the flower development and senescence of Rhododendron delavayi. Overall, the manuscript is well written and the results presented are appealing. Nevertheless, some corrections can be made on the lines mentioned below.
- The abstract must include data regarding the critical findings by the authors in terms of data of important findings.
- In this manuscript's introduction, there needs to be a clear hypothesis and a lot of development of the second paragraph.
- Overall, there is a repetition of the information which could be avoided.
- Check the figures ligands; they are carelessly written.
- Discussion should include more information and references related to the relevant and related works.
- Restructure and carefully edit the conclusion section.
Author Response

(The authors gave the same response as above.)

Round 2
Reviewer 1 Report
Seems there are no figures shown in the revised manuscript.
Author Response
Dear reviewer:
Thank you for your comment again on our manuscript entitled “Identification and Characterization of Circular RNAs Involved in the Flower Development and Senescence of Rhododendron delavayi Franch” (ijms-1893749). According to your suggestion, we have made amendments in the revised manuscript carefully. Here, we respond to your comment as follows. Please see the attachment.
Best regards,
Jing Tang, PhD
